# Two-stage Auction Design in Online Advertising

## Abstract

Modern online advertising systems usually involve a large amount of advertisers in each auction, causing scalability issues. To mitigate the problem, two-stage auctions are designed and deployed in practice, enabling efficient allocations of ad slots among numerous candidate advertisers within a short response time. Such a design uses a fast but coarse model to select a small subset of advertisers in the first stage, and a slow yet refined model to finally decide the winners. However, existing two-stage auction mechanisms primarily focus on optimizing welfare, ignoring other crucial objectives of the platform, such as revenue.

In this paper, we propose ad-wise selection metrics (namely Max-Wel and Max-Rev) that are based on an ad's contribution to the platform's objective (welfare or revenue). Then we provide theoretical guarantees for the proposed metrics. Our method is applicable to both welfare and revenue optimizations and can be easily implemented using neural networks. We conduct extensive experiments on both synthetic and industrial data to demonstrate the advantages of our proposed selection metrics over existing baselines.

## CCS Concepts

• **Theory of computation → Algorithmic game theory and mechanism design**; • **Computing methodologies** → *Neural networks*.

## Keywords

Mechanism design, Online advertising, Neural networks

**ACM Reference Format:**

Anonymous Author(s). 2018. Two-stage Auction Design in Online Advertising. In *Proceedings of Make sure to enter the correct conference title from your rights confirmation emai (Conference acronym 'XX)*. ACM, New York, NY, USA, 13 pages. https://doi.org/XXXXXXX.XXXXXXX

## 1 Introduction

Online advertising plays a vital role in modern Internet companies and serves as their primary source of revenue [9]. In modern advertising systems, when a user makes a request, the platform allocates multiple ad slots to candidate advertisers through ad auctions [9, 25]. Since these auctions are in real-time, the final allocation of ad slots must be determined within tens of milliseconds [12]. Moreover, to ensure efficient and effective allocation, auction outcomes often rely not only on the advertisers' bids but also on indicators of the

relevance of the ad to the current user, such as click-through rate (CTR) and conversion rate (CVR), which we collectively refer to as the ad quality. In real-world scenarios, the platform often utilizes a refined but heavy machine learning model to predict the quality of each ad [16, 28]. However, with the growing number of candidate advertisers, this heavy model can only be applied to a subset of the entire ad set due to time constraints.

To address this scalability issue, the platform turns to use a two-stage auction architecture: in the first stage, it swiftly selects a small subset of ads to enter the next stage using a lightweight but coarse ML model; while in the second stage, it employs a refined ML model on the remaining advertisers to determine the final auction outcome. In recent years, the two-stage auction design problem has attracted attention from numerous researchers and can be broadly categorized into two lines. One line of research focuses on an underlying optimization problem: given rough estimates of advertiser quality, such as its distribution, how to select a subset of advertisers to maximize the objective, also known as the bidder selection problem [2, 11, 20, 23]. However, the strong and unrealistic assumption about known distribution information makes it challenging to apply these algorithms to real-world scenarios. Another line of research studies this problem from a machine learning perspective [26]: assuming that only partial features can be used in the first stage, how to rapidly and efficiently select high-quality advertisers to proceed to the second stage by a machine learning model. Our main focus in this paper lies in the latter. Existing work in this line mainly falls short in the following aspects: 1) Due to the difficulties involved, most research addresses a related, albeit different, problem rather than directly tackling the original optimization problem. 2) Much of the existing work lacks theoretic foundations and guarantees. 3) only the welfare is considered as the optimization objective while overlooking other important goals of the platform including the revenue.

To address the aforementioned limitations, we propose novel selection metrics for the advertiser selection problem. First, we formulate the two-stage auction as an optimization problem with both welfare and revenue as the objectives. We derive ad-wise selection metrics based on theoretic analysis of the auctions. We rank the ads according to their expected contributions to the objective functions, where the top $m$ ads are selected to proceed to the next stage. Since the proposed metrics are based on auction theory, we are able to provide approximation bounds for each metric under different degrees of assumptions. We also design a learning-based implementation of our method, which can be trained using existing auction data. Finally, we conduct extensive experiments to demonstrate the effectiveness of our proposed method. Specifically, we compare the performance of our proposed methods and existing two-stage baselines in terms of both welfare and revenue, using both synthetic and industrial data. We find that our proposed method consistently outperforms baselines. Moreover, we compare the performance of different methods across different selection sizes and find that the margin of improvement tends to be larger when the number of

advertisers selected in the first stage is smaller. This indicates that our metrics excel in prioritizing high-quality advertisers.

## 1.1 Related Work

Learning-based auction design [8, 13, 24] has received considerable attention recently, particularly in the online advertising scenario [17, 19]. However, to our knowledge, there is limited research on two-stage auctions. One of the most relevant works to ours is the study by Wang et al. [26], who propose a selection metric for welfare maximization by maximizing the expected recall. In contrast, our approach considers both welfare and revenue maximization, using each ad's contribution to the objective as the selection metric. We also provide theoretical guarantees for our method.

Another related topic is the subset selection problem under uncertainty, which has been investigated across various scenarios, including search engine [4], voting theory [21], team selection [15], and procurement auctions [22]. In the realm of online advertising, this problem is often referred to as the bidder selection problem (BSP). Previous work by Chen et al. [5] can be viewed as the BSP for maximizing welfare in a VCG setting. Mehta et al. [20] extend this by considering both welfare and revenue maximization. Bei et al. [2] studies the BSP for maximizing revenue under multiple auction formats with a single item. These studies assume knowledge of the distribution of participants' values. In contrast, our focus in this paper is to leverage data-driven advantages to aid bidder selection from a machine-learning perspective.

## 2 Preliminaries

We consider the two-stage auction problem for an online advertising platform (e.g., a search engine). When a user of such a platform performs a specific action (for example, entering a query in a search engine), the platform displays several ads along with the organic content. The space that contains the ads is called the slots, and these slots are usually sold through auctions. Once an ad auction is triggered, the platform asks the advertisers to submit their bids and then decides the winners based on their bids.

Throughout this paper, we assume that there are $n$ potential advertisers competing for $K$ slots and each advertiser has only one ad. We sometimes use "ad" and "advertiser" interchangeably. Denote by $N = \{1, \cdots, n\}$ the set of all possible advertisers. Each advertiser $i \in N$ has a private value $v_i \in \mathbb{R}_+$ that captures their payoff for an ad click. Based on their private value, each advertiser submits a bid $b_i \in \mathbb{R}_+$. We use $\boldsymbol{b} = (b_1, \ldots, b_n)$ to represent the bid vector of all advertisers. Aside from $\boldsymbol{b}$, the ad auction results also depend on the quality of the ad itself, which we denote by $q_i$. The quality of an ad can reflect the current user's interest in the ad. Similar to most existing works, we use the click-through rate (CTR) as our quality index, i.e., $q_i$ is the probability of the user clicking on the ad. We use $\boldsymbol{q} = (q_1, q_2, \ldots, q_n)$ to denote the CTR profile of all advertisers for the current user.

An ad auction mechanism consists of two components: an allocation rule $x = (x_1, x_2, \ldots, x_n)$ and a payment rule $p = (p_1, p_2, \ldots, p_n)$. The allocation rule $x_i(\boldsymbol{b}, \boldsymbol{q})$ is a function that outputs an integer indicating which slot is allocated to advertiser $i$, i.e., $x_i(\boldsymbol{b}, \boldsymbol{q}) = j$ represents that advertiser $i$ wins the $j$-th slot, and $j = 0$ means that the advertiser loses this auction. The payment rule $p_i(\boldsymbol{b}, \boldsymbol{q})$ is also

a function that outputs a real number representing the fee that the advertiser $i$ needs to pay when his ad is clicked by the user. When we refer to $K$ slots, it indicates that a total of $K$ winners will win the auction. In this paper, we view $K$ as a given constant and consider one of the most widely used auction mechanisms: the generalized second-price auction (GSP). The GSP auction first ranks all the ads by a score $s_i = b_i q_i$, and then allocates the $j$-th slot to the $j$-th highest advertiser. If the $j$-th ad is clicked by the user, he pays the least amount that would retain his slot $j$. Formally:

$$x_i(\boldsymbol{b}, \boldsymbol{q}) = \begin{cases} j & \text{if } s_i = s_{(j)} \\ 0 & \text{otherwise} \end{cases}, \quad p_i(\boldsymbol{b}, \boldsymbol{q}) = \begin{cases} \frac{s_{(j+1)}}{q_i} & \text{if } x_i(\boldsymbol{b}, \boldsymbol{q}) = j \\ 0 & \text{otherwise} \end{cases},$$

(1)

where the subscript $(j)$ refers to the advertiser that has the $j$-th highest score.

## 2.1 CTR Prediction and Two-stage Auctions

In real-world applications, the CTR of an ad is usually predicted by machine learning models. Therefore, the performance of the ad auction depends not only on the mechanism itself but also on the accuracy of the ad CTR estimator. In the last decade, a variety of learning models have been proposed to estimate the CTR of ads relative to the user, and different learning models may use different inputs.

A naive and straightforward two-stage auction usually makes use of two CTR models: a lightweight but coarse model $\mathcal{M}^c$ and a heavy but refined model $\mathcal{M}^r$. In the first stage, it uses the coarse model $\mathcal{M}^c$ to select potential winners to enter the next stage. And in the second stage, it uses the refined model $\mathcal{M}^r$ to determine the final winners. Compared with the refined model, the coarse model uses fewer features and thus is computationally more efficient but less accurate. Formally, let $a_i$ and $u$ be the features of ad $i$ and the user that is used by the refined model. We sometimes call them the full features. The coarse model only uses partial features (i.e., a subset of full features) which we denote by $\tilde{a}_i$ and $\tilde{u}$, respectively. Therefore, the CTRs predicted by the two models are

$$q_i = \mathcal{M}^r(a_i, u), \quad \tilde{q}_i = \mathcal{M}^c(\tilde{a}_i, \tilde{u}). \tag{2}$$

We assume that two models are trained using the same set of data $\mathcal{D}$. Let $\mathcal{D}(\star, \diamond) \subseteq \mathcal{D}$ be the set of data that contains feature $(\star, \diamond)$. As mentioned in [26], a well-trained CTR model should satisfy:

$$q_i = \frac{|D^+(a_i, u)|}{|D(a_i, u)|}, \quad \tilde{q}_i = \sum_{a_i|\tilde{a}_i, u|\tilde{u}} q_i \times Pr[a_i, u|\tilde{a}_i, \tilde{u}] = \underset{a_i|\tilde{a}_i, u|\tilde{u}}{\mathbf{E}}[q_i],$$

where $\mathcal{D}^+$ is the set of positive data, i.e., the clicked data. Obviously, the naive two-stage mechanism fails to consider the relationship between the two CTR estimators, rendering it an ineffective solution. Next, we delve into the two-stage auction design problems based on this relationship.

In line with prior works [11, 26], we adopt the GSP mechanism for the second stage. Consequently, our primary focus lies in the design of the first stage, where we are constrained to use only partial features to select a subset of advertisers. We consider two kinds of objectives commonly used in the literature: the welfare and the revenue. The welfare of an auction is defined as the total

value[1] realized through the auction. Formally, the welfare can be written as follows:

$$\text{Wel} = \sum_{i \in N} b_i q_i \mathbb{I}\left\{x_i(\boldsymbol{b}, \boldsymbol{q}) > 0\right\},$$

where $\mathbb{I}\{\cdot\}$ is the indicator function. The revenue of an auction is the total payment received by the platform. Under a GSP auction, the revenue can be written as:

$$\text{Rev} = \sum_{i \in N} p_i q_i \mathbb{I}\left\{x_i(\boldsymbol{b}, \boldsymbol{q}) > 0\right\}.$$

The objective of the first stage is to select a subset $M \subseteq N$ of ads with size $|M| = m \geq K$ to enter the second stage such that the objective is maximized, given partial feature $(\tilde{\boldsymbol{a}}, \tilde{u})$ and bid profile $\boldsymbol{b}$. Denote by $Top_M^K(\boldsymbol{b}, \boldsymbol{q})$ the set of $K$ ads with the highest $s_i$ in subset $M$. Thus the optimization problem in the first stage can be phrased as:

$$\max_{M \subseteq N} \operatorname*{\mathbb{E}}_{\boldsymbol{a}|\tilde{\boldsymbol{a}}, u|\tilde{u}} \left[\sum_{i \in M} b_i q_i \mathbb{I}\left\{i \in Top_M^K(\boldsymbol{b}, \boldsymbol{q})\right\}\right] \quad \text{or}$$

$$\max_{M \subseteq N} \operatorname*{\mathbb{E}}_{\boldsymbol{a}|\tilde{\boldsymbol{a}}, u|\tilde{u}} \left[\sum_{i \in M} p_i q_i \mathbb{I}\left\{i \in Top_M^K(\boldsymbol{b}, \boldsymbol{q})\right\}\right],$$

depending on whether the final goal of the platform is to maximize the welfare or the revenue. In the above optimization problems, $q_i$ is computed by $\mathcal{M}^r$ in Equation (2), and $\tilde{\boldsymbol{a}} = (\tilde{a}_1, \tilde{a}_2, \ldots, \tilde{a}_n)$, $\boldsymbol{a} = (a_1, a_2, \ldots, a_n)$ are the profiles of the partial and full features of all the ads, respectively.

*Incentive Issues.* We also need to take into account the advertisers' incentives to misreport their values when designing a two-stage auction. *Incentive compatibility* (or simply IC) is one of the most important economic properties in auction design. An auction mechanism is IC if it is in their best interest to report their true valuations. Fortunately, it is known from [27] that for value-maximizing advertisers, truthfully reporting is the best strategy if the mechanism satisfies the following conditions: (i) Monotonicity: an advertiser would win the same ad slot or a higher one if she reports a higher bid. (ii) Critical price: the payment for a winning advertiser is the minimum bid that she needs to maintain the same ad slot. We assume all advertisers are value maximizers since this model captures the goals of most advertisers in the advertising scenario. The GSP auction already satisfies these two conditions, and the first stage has nothing to do with payment. Therefore, to ensure the IC property of a two-stage mechanism, we only need to ensure the monotonicity of allocation in the first stage.

## 3 First-stage Ad Selection Metric for Welfare Maximization

Intuitively, the objective in the first stage is to select as many "good" ads as possible from the whole ad set $N$. An ad can be regarded as a "good" ad if the inclusion of the ad makes a fair amount of contribution to the welfare of the auction. The following theorem gives

---

[1]A more rigorous definition of the social welfare should use the advertisers' values $v$ instead of their bids $b$. However, according to Wilkens et al. [27], value-maximizing advertisers use the strategy $b_i = v_i$ in the GSP auction. Thus, we do not distinguish between the value $v_i$ and the bid $b_i$ and just use $b_i$ hereafter.

the contribution of each advertiser to the welfare of the platform. For simplicity, define:

$$\text{Wel}(M|\boldsymbol{b}, \tilde{\boldsymbol{a}}, \tilde{u}) = \operatorname*{\mathbb{E}}_{\boldsymbol{a}|\tilde{\boldsymbol{a}}, u|\tilde{u}} \left[\sum_{i \in M} b_i q_i \mathbb{I}\left\{i \in Top_M^K(\boldsymbol{b}, \boldsymbol{q})\right\}\right].$$

THEOREM 1. *Given any bid profile $\boldsymbol{b}$ and partial feature $\tilde{\boldsymbol{a}}, \tilde{u}$, the expected contribution of advertiser $i$ to the welfare objective function is:*

$$f_i(\boldsymbol{b}, \tilde{\boldsymbol{a}}, \tilde{u}) = \operatorname*{\mathbb{E}}_{q_i|\tilde{q}_i} \left[b_i q_i \Pr\left\{i \in Top_N^K(\boldsymbol{b}, \boldsymbol{q}) \,\middle|\, \boldsymbol{b}, \tilde{\boldsymbol{q}}\right\}\right]. \quad (3)$$

PROOF. If we can select a subset $M$ with a size equal to $n$ (i.e., set $M = N$), then all ads can enter into the second stage. This essentially reduces the problem to single-stage auctions, thereby achieving optimal welfare. Then we have:

$$\max_{M \subseteq N} \text{Wel}(M|\boldsymbol{b}, \tilde{\boldsymbol{a}}, \tilde{u}) \leq \mathbb{E}_{\boldsymbol{a}|\tilde{\boldsymbol{a}}, u|\tilde{u}} \left[\sum_{i \in N} b_i q_i \mathbb{I}\left\{i \in Top_N^K(\boldsymbol{b}, \boldsymbol{q})\right\}\right].$$

The right-hand side of the above equation can also be written as:

$$\operatorname*{\mathbb{E}}_{\boldsymbol{a}|\tilde{\boldsymbol{a}}, u|\tilde{u}} \left[\sum_{i \in N} b_i q_i \mathbb{I}\left\{i \in Top_N^K(\boldsymbol{b}, \boldsymbol{q})\right\}\right]$$

$$= \sum_{i \in N} \operatorname*{\mathbb{E}}_{\boldsymbol{q}|\tilde{\boldsymbol{q}}} \left[b_i q_i \mathbb{I}\left\{i \in Top_N^K(\boldsymbol{b}, \boldsymbol{q})\right\}\right]$$

$$= \sum_{i \in N} \operatorname*{\mathbb{E}}_{q_i|\tilde{q}_i} \left[b_i q_i \mathbb{E}_{\boldsymbol{q}_{-i}|\tilde{\boldsymbol{q}}_{-i}} \left[\mathbb{I}\left\{i \in Top_N^K(\boldsymbol{b}, \boldsymbol{q})\right\}\right]\right]$$

$$= \sum_{i \in N} \operatorname*{\mathbb{E}}_{q_i|\tilde{q}_i} \left[b_i q_i \Pr\left\{i \in Top_N^K(\boldsymbol{b}, \boldsymbol{q}) \middle| \boldsymbol{b}, \tilde{\boldsymbol{q}}\right\}\right].$$

Define function $f_i(\boldsymbol{b}, \tilde{\boldsymbol{a}}, \tilde{u})$ as follows:

$$f_i(\boldsymbol{b}, \tilde{\boldsymbol{a}}, \tilde{u}) = \operatorname*{\mathbb{E}}_{q_i|\tilde{q}_i} \left[b_i q_i \Pr\left\{i \in Top_N^K(\boldsymbol{b}, \boldsymbol{q}) \,\middle|\, \boldsymbol{b}, \tilde{\boldsymbol{q}}\right\}\right].$$

Then the right-hand side of the above equation is a summation of $f_i(\boldsymbol{b}, \tilde{\boldsymbol{a}}, \tilde{u})$ over all ads. Therefore, the ranking index $f_i(\boldsymbol{b}, \tilde{\boldsymbol{a}}, \tilde{u})$ can be viewed as the expected contribution of ad $i$ to the objective. In fact, if ad $i$ is a winner, its contribution is $b_i q_i$ by definition. Thus, $f_i(\boldsymbol{b}, \tilde{\boldsymbol{a}}, \tilde{u})$ is indeed the expected contribution of ad $i$ to the objective and can serve as an ad-wise selection metric for the bidder selection problem in the first stage. □

To maximize welfare, we can rank ads according to their expected contributions and select top $m$ ads to proceed to the next stage. Thus $M$ is in fact a set-valued function with input $\boldsymbol{b}, \tilde{\boldsymbol{a}}, \tilde{u}$. We show that the expected welfare contribution of set $M$ is a lower bound of the actual welfare of set $M$, that is:

$$\text{Wel}(M|\boldsymbol{b}, \tilde{\boldsymbol{a}}, \tilde{u}) \geq \sum_{i \in M} f_i(\boldsymbol{b}, \tilde{\boldsymbol{a}}, \tilde{u}).$$

LEMMA 1. *For any selected set $M \subseteq N$, the expected welfare contribution of set $M$ is a lower bound of the actual welfare of choosing set $M$.*

In real-world advertising systems, the ranking index $f_i(\boldsymbol{b}, \tilde{\boldsymbol{a}}, \tilde{u})$ can be approximated by a neural network. However, the input of the network includes the bids and features of all participating advertisers. The dimension of the input can be very high and even variable, making it difficult to design and train the network. In fact,

a typical online ad platform can have more than 100,000 advertisers. The set of active advertisers may be different for different users and the benefit of including the feature of inactive advertisers may not be able to compensate for the difficulties posed by them in the training of the network. Therefore, from now on, we will consider the following simplified version of $f_i(\boldsymbol{b}, \tilde{a}, \tilde{u})$ that only depends on the features of ad $i$ itself:

$$\bar{f}_i(b_i, \tilde{a}_i, \tilde{u}) = \mathop{\mathrm{E}}_{\boldsymbol{b}_{-i}, \tilde{a}_{-i}} [f_i(\boldsymbol{b}, \tilde{a}, \tilde{u})].$$

## 3.1 Ranking Score Monotonicity

Recall that the refined CTR $q_i$ is a random variable with mean $\tilde{q}_i$. As a result, the score $s_i = b_i q_i$ is also a random variable. Suppose that all the random scores are independently conditioned on the bid profile $\boldsymbol{b}$ and the partial features $\tilde{a}$ and $\tilde{u}$. We show that the expected contribution $\bar{f}_i(b_i, \tilde{a}_i, \tilde{u})$ of ad $i$ is larger than that of ad $j$, if the score $s_i$ of ad $i$ stochastically dominates the score $s_j$ of ad $j$, where the relation of stochastic dominance is defined as follows:

DEFINITION 1 (STOCHASTIC DOMINANCE). *Random variable $X$ stochastically dominates random variable $Y$, if $F_X(t) \leq F_Y(t), \forall t$, where $F_X(t)$ and $F_Y(t)$ are the cumulative distribution functions of $X$ and $Y$, respectively.*

*An alternative and equivalent definition is that $X$ stochastically dominates $Y$, if $\mathrm{E}_X[u(X)] \geq \mathrm{E}_Y[u(Y)]$ for any increasing function $u : \mathbb{R} \mapsto \mathbb{R}$.*

Denote by $G_i(t)$ and $G_j(t)$ the cumulative distribution functions of the random scores $s_i$ and $s_j$, and by $g_i(t)$ and $g_i(t)$ their corresponding density function. Here, we assume that all scores are continuous random variables to avoid the complication of point masses and tie-breaking rules. Formally, we have the following result.

THEOREM 2. *Given any bid profile $\boldsymbol{b}$ and partial feature $\tilde{a}$ and $\tilde{u}$, if the resulting random scores are independent and $s_i$ stochastically dominates $s_j$, then $\bar{f}_i(b_i, \tilde{a}_i, \tilde{u}) \geq \bar{f}_j(b_j, \tilde{a}_j, \tilde{u})$.*

PROOF. Fix $s_l$ for all $l \in N$ with $l \neq i, j$. For ease of presentation, denote by $c_k$ the $k$-th largest score in the set $\{s_l \mid l \in N, l \neq i, j\}$. Clearly, we have $c_{K-1} \geq c_K$. Now consider the following three cases.

**Case 1.** $s_i \geq c_{K-1}$. In this case, no matter what the actual value of $s_j$ is, ad $i$ is always among the top $K$ ads, i.e., $i \in Top_N^K(\boldsymbol{b}, \boldsymbol{q})$. Therefore, the contribution of ad $i$ to the welfare is simply $s_i$, and the expected contribution of ad $i$ in this case is:

$$\int_{c_{K-1}}^{\infty} s_i g_i(s_i) s_i.$$

**Case 2.** $c_K \leq s_i < c_{K-1}$. In this case, there are already $K - 1$ ads with scores higher than $s_i$. So ad $i$ can contribute to the welfare only if $s_j \leq s_i$. Since $s_j$ and $s_i$ are independent given $\boldsymbol{b}, \tilde{a}$, and $\tilde{u}$, $s_j \leq s_i$ happens with probability $G_j(s_i)$. Therefore, the total contribution of ad $i$ in this case is:

$$\int_{c_K}^{c_{K-1}} s_i g_i(s_i) G_j(s_i) s_i.$$

**Case 3.** $s_i < c_K$. In this case, we already have $K$ ads with scores exceeding $s_i$. Thus, ad $i$ cannot make a non-zero contribution even if $s_j < s_i$. So the total contribution is simply 0.

Combining the contributions in the three cases together, the total contribution of ad $i$ is:

$$\int_{c_{K-1}}^{\infty} s_i g_i(s_i) s_i + \int_{c_K}^{c_{K-1}} s_i g_i(s_i) G_j(s_i) s_i.$$

Define:

$$\tilde{G}_j(s_i) = \begin{cases} 0 & \text{if } 0 \leq s_i \leq c_K \\ s_i G_j(s_i) & \text{if } c_K < s_i \leq c_{K-1} , \\ s_i & \text{if } s_i > c_{K-1} \end{cases}$$

and

$$\tilde{G}_i(s_j) = \begin{cases} 0 & \text{if } 0 \leq s_j \leq c_K \\ s_j G_i(s_j) & \text{if } c_K < s_j \leq c_{K-1} . \\ s_j & \text{if } s_j > c_{K-1} \end{cases}$$

Then the total contribution of ad $i$ can be re-written as:

$$\int_0^{\infty} \tilde{G}_j(s_i) g_i(s_i) s_i.$$

Similarly, the contribution of ad $j$ can be obtained by switching the role of $i$ and $j$:

$$\int_0^{\infty} \tilde{G}_i(s_j) g_j(s_j) s_j.$$

Since $s_i$ stochastically dominates $s_j$, by definition, we have $G_i(t) \leq G_j(t), \forall t$, which implies $\tilde{G}_j(t) \geq \tilde{G}_i(t), \forall t$. Consequently,

$$\begin{aligned} \mathop{\mathrm{E}}_{s_i} \left[ \tilde{G}_j(s_i) \right] &= \int_0^{\infty} \tilde{G}_j(t) g_i(t) t \\ &\geq \int_0^{\infty} \tilde{G}_i(t) g_i(t) t \\ &= \mathop{\mathrm{E}}_{s_i} \left[ \tilde{G}_i(s_i) \right] \\ &\geq \mathop{\mathrm{E}}_{s_j} \left[ \tilde{G}_i(s_j) \right], \end{aligned}$$

where the last inequality is due to the alternative definition of stochastic dominance.

Through the above analysis, we know that the contribution of ad $i$ is always larger than that of ad $j$ for any fixed scores of other ads. Taking expectation over the scores of other ads immediately leads to the conclusion that the expected contribution of ad $i$ to the welfare is larger than that of ad $j$, or equivalently, $\bar{f}_i(b_i, \tilde{a}_i, \tilde{u}) \geq \bar{f}_j(b_j, \tilde{a}_j, \tilde{u})$. □

## 3.2 Welfare Approximation

We derive approximation results for the selection metric. Based on the approximation results, we are able to calculate the size of $\bar{M}$ needed to guarantee a certain fraction of the optimal welfare.

The above analyses are based on any given bid $\boldsymbol{b}$ and partial features $\tilde{a}$ and $\tilde{u}$. The total expected welfare of the platform can be obtained by taking expectations over them. To analyze the performance of the ranking index $\bar{f}_i(b_i, \tilde{a}_i, \tilde{u})$, we further assume that bid $b_i$ and partial feature $\tilde{a}_i$ are independent across the ads. Thus the ranking index $\bar{f}_i(b_i, \tilde{a}_i, \tilde{u})$ is also a random variable for any user feature $\tilde{u}$ and is also independent across all ads.

### 3.2.1 Uniform Distribution.
We start with the simplest case where the ranking indices $\bar{f}_i$ of all ads are i.i.d. random variables that follow the uniform distribution over the interval $[a, b]$.

**LEMMA 2.** *Suppose the ranking indices $\bar{f}_i$ are i.i.d random variables that follow uniform distribution $U[a, b]$ with $0 \le a < b$. Then always including the ads with top $m$ ranking indices achieves an $\alpha$ fraction of the optimal welfare if:*

$$m > \frac{b}{2(b-a)} \left[ 2n + 2 - \sqrt{1-\alpha}(2n+1) \right].$$

### 3.2.2 General Distribution.
In addition to uniform distributions, we also give results for general distributions with mean $\mu$ and variance $\sigma^2$. We still assume that the ranking indices are i.i.d. random variables. Our welfare approximation result for general distributions is as follows.

**LEMMA 3.** *Suppose the ranking indices $\bar{f}_i$ are random variables that follow a distribution with mean $\mu$ and variance $\sigma^2$. Then always including the ads with top $m$ ranking indices achieves an $\alpha$ fraction of the optimal welfare if:*

$$m > \alpha n - \frac{\sigma}{2\mu}n + \frac{\sqrt{2}}{4}(2n+1)\sqrt{\frac{\sigma}{\mu}}.$$

## 4 First-stage Ad Selection Metric for Revenue Maximization

In revenue-maximizing scenarios, the objective of the first stage is also to select advertisers who contribute more to revenue. Here, the revenue contribution of an ad is the expected payment made by the advertiser.

Firstly, we give the equivalent form of the objective function of revenue.

**LEMMA 4.** *The objective function for maximizing revenue can be equivalently expressed as:*

$$\max_{M \subseteq N} \mathop{\mathbf{E}}_{a|\tilde{a},u|\tilde{u}} \left[ \sum_{i \in M} s_i \mathbb{I}\left\{ i \in Top_M^{K+1}(\boldsymbol{b}, \boldsymbol{q}) \right\} - \sum_{i \in M} s_i \mathbb{I}\left\{ i \in Top_M^1(\boldsymbol{b}, \boldsymbol{q}) \right\} \right]. \tag{4}$$

**PROOF.** Recall that the revenue optimization problem in the first stage can be phrased as:

$$\max_{M \subseteq N} \mathop{\mathbf{E}}_{a|\tilde{a},u|\tilde{u}} \left[ \sum_{i \in M} p_i q_i \mathbb{I}\left\{ i \in Top_M^K(\boldsymbol{b}, \boldsymbol{q}) \right\} \right].$$

Combined with the definition of payment function $p_i$, the optimization problem can be rewritten as:

$$\max_{M \subseteq N} \mathop{\mathbf{E}}_{a|\tilde{a},u|\tilde{u}} \left[ \sum_{i \in M} \sum_{j=1}^{K} s_M^{(j+1)} \mathbb{I}\left\{ s_i = s_M^{(j)} \right\} \right],$$

where $s_M^{(j)}$ denotes the $j$-th highest score in the ad set $M$. Note that given $\boldsymbol{a}$ and $u$, we have:

$$\sum_{i \in M} \sum_{j=1}^{K} s_M^{(j+1)} \mathbb{I}\left\{ s_i = s_M^{(j)} \right\} = \sum_{j=1}^{K} s_M^{(j+1)}$$

$$= \sum_{j=1}^{K+1} s_M^{(j)} - s_M^{(1)}$$

$$= \sum_{i \in M} s_i \mathbb{I}\left\{ i \in Top_M^{K+1} \right\} - \sum_{i \in M} s_i \mathbb{I}\left\{ i \in Top_M^1 \right\}$$

Then we obtain an alternative objective:

$$\max_{M \subseteq N} \mathop{\mathbf{E}}_{a|\tilde{a},u|\tilde{u}} \left[ \sum_{i \in M} s_i \mathbb{I}\left\{ i \in Top_M^{K+1}(\boldsymbol{b}, \boldsymbol{q}) \right\} - \sum_{i \in M} s_i \mathbb{I}\left\{ i \in Top_M^1(\boldsymbol{b}, \boldsymbol{q}) \right\} \right].$$

$\square$

Then we derive each ad's contribution to the revenue objective. For simplicity, we define:

$$\text{REV}(M|\boldsymbol{b}, \tilde{a}, \tilde{u}) = \mathop{\mathbf{E}}_{a|\tilde{a},u|\tilde{u}} \left[ \sum_{i \in M} s_i \mathbb{I}\left\{ i \in Top_M^{K+1}(\boldsymbol{b}, \boldsymbol{q}) \right\} \right.$$

$$\left. - \sum_{i \in M} s_i \mathbb{I}\left\{ i \in Top_M^1(\boldsymbol{b}, \boldsymbol{q}) \right\} \right].$$

**THEOREM 3.** *Given any bid profile $\boldsymbol{b}$ and partial feature $\tilde{a}, \tilde{u}$, the expected contribution of advertiser $i$ to the revenue objective function is:*

$$r_i(\boldsymbol{b}, \tilde{a}, \tilde{u}) = \mathop{\mathbf{E}}_{q_i|\tilde{q}_i} \left[ s_i \Pr\left\{ i \in Top_N^{K+1}(\boldsymbol{b}, \boldsymbol{q}) \Big| \boldsymbol{b}, \tilde{\boldsymbol{q}} \right\} \right]$$

$$- \mathop{\mathbf{E}}_{q_i|\tilde{q}_i} \left[ s_i \Pr\left\{ i \in Top_N^1(\boldsymbol{b}, \boldsymbol{q}) \Big| \boldsymbol{b}, \tilde{\boldsymbol{q}} \right\} \right]. \tag{5}$$

**PROOF.** If we can select a subset $M$ with a size equal to $n$ (i.e., set $M = N$), then all ads can enter into the second stage. This essentially reduces the problem to single-stage auctions, thereby achieving optimal revenue. Then we have:

$$\max_{M \subseteq N} \text{REV}(M|\boldsymbol{b}, \tilde{a}, \tilde{u}) \le \mathop{\mathbf{E}}_{a|\tilde{a},u|\tilde{u}} \left[ \sum_{i \in N} s_i \mathbb{I}\left\{ i \in Top_N^{K+1}(\boldsymbol{b}, \boldsymbol{q}) \right\} \right.$$

$$\left. - \sum_{i \in N} s_i \mathbb{I}\left\{ i \in Top_N^1(\boldsymbol{b}, \boldsymbol{q}) \right\} \right]. \tag{6}$$

The right-hand side of the above inequality can be written as:

$$\mathop{\mathbf{E}}_{a|\tilde{a},u|\tilde{u}} \left[ \sum_{i \in N} s_i \mathbb{I}\left\{ i \in Top_N^{K+1}(\boldsymbol{b}, \boldsymbol{q}) \right\} - \sum_{i \in N} s_i \mathbb{I}\left\{ i \in Top_N^1(\boldsymbol{b}, \boldsymbol{q}) \right\} \right]$$

$$= \sum_{i \in N} \mathop{\mathbf{E}}_{q|\tilde{q}} \left[ s_i \mathbb{I}\left\{ i \in Top_N^{K+1}(\boldsymbol{b}, \boldsymbol{q}) \right\} \right] - \sum_{i \in N} \mathop{\mathbf{E}}_{q|\tilde{q}} \left[ s_i \mathbb{I}\left\{ i \in Top_N^1(\boldsymbol{b}, \boldsymbol{q}) \right\} \right]$$

$$= \sum_{i \in N} \mathop{\mathbf{E}}_{q_i|\tilde{q}_i} \left[ s_i \Pr\left\{ i \in Top_N^{K+1}(\boldsymbol{b}, \boldsymbol{q}) \Big| \boldsymbol{b}, \tilde{\boldsymbol{q}} \right\} \right]$$

$$- \sum_{i \in N} \mathop{\mathbf{E}}_{q_i|\tilde{q}_i} \left[ s_i \Pr\left\{ i \in Top_N^1(\boldsymbol{b}, \boldsymbol{q}) \Big| \boldsymbol{b}, \tilde{\boldsymbol{q}} \right\} \right].$$

We define function $r_i(b, \tilde{a}, \tilde{u})$ as follows:

$$r_i(b, \tilde{a}, \tilde{u}) = \underset{q_i|\tilde{q}_i}{\mathrm{E}} \left[ s_i \Pr \left\{ i \in Top_N^{K+1}(b, q) \middle| b, \tilde{q} \right\} \right]$$
$$- \underset{q_i|\tilde{q}_i}{\mathrm{E}} \left[ s_i \Pr \left\{ i \in Top_N^1(b, q) \middle| b, \tilde{q} \right\} \right]$$
$$= f_i^{(K+1)}(b, \tilde{a}, \tilde{u}) - f_i^{(1)}(b, \tilde{a}, \tilde{u}).$$

Then the right-hand side of the inequality (6) is a summation of $r_i(b, \tilde{a}, \tilde{u})$ over all ads. Then we can regard $r_i(b, \tilde{a}, \tilde{u})$ as the expected revenue contribution of ad $i$.                                    □

Note that, combined with Equation (3), Equation (5) can also be written as:

$$r_i(b, \tilde{a}, \tilde{u}) = f_i^{(K+1)}(b, \tilde{a}, \tilde{u}) - f_i^{(1)}(b, \tilde{a}, \tilde{u}),$$

where the superscript $(K + 1)$ denotes the number of ad slots.

## 4.1 Refined Selection Metric for Revenue Maximizing

If there is a bidder who is significantly superior to other bidders (for example, both bid and ctr are high), then the probability that he is in $Top_N^{K+1}$ is very close to the probability that he is in $Top_N^1$ (both close to 1). However, in this case, the revenue contribution of the bidder calculated by $r_i$ is very low, which is obviously unreasonable. In general second price auction, without the highest bidder, all the other bidder's payments have to go down. This happens because there is a subtraction in the revenue contribution $r_i$. To address this issue, we define a surrogate ranking index as follows:

$$R_i(b, \tilde{a}, \tilde{u}) = f_i^{(K+1)}(b, \tilde{a}, \tilde{u}),$$

which is equivalent to the bidder $i$'s welfare contribution when there is $K + 1$ slots. Then we determine the candidate set $M$ by selecting the ads with the highest refined revenue ranking indices $R_i$. The actual revenue of choosing set $M$ is:

$$\mathrm{Rev}(M|b, \tilde{a}, \tilde{u}) = \sum_{i \in M} \underset{q|\tilde{q}}{\mathrm{E}} \left[ s_i \mathbb{I} \left\{ i \in Top_M^{K+1}(b, q) \right\} \right]$$
$$- \sum_{i \in M} \underset{q|\tilde{q}}{\mathrm{E}} \left[ s_i \mathbb{I} \left\{ i \in Top_M^1(b, q) \right\} \right].$$

Next, we show that $\mathrm{Rev}(M|b, \tilde{a}, \tilde{u}) \geq \sum_{i \in M} r_i(b, \tilde{a}, \tilde{u})$.

LEMMA 5. *For any selected $M \subseteq N$, the expected revenue contribution of $M$ is a lower bound of the actual revenue of choosing $M$.*

Here, we also define the simplified ranking index and simplified refined index that only depends on the features of ad $i$ as follows:

$$\bar{r}_i(b_i, \tilde{a}_i, \tilde{u}) = \bar{f}_i^{(K+1)}(b_i, \tilde{a}_i, \tilde{u}) - \bar{f}_i^{(1)}(b_i, \tilde{a}_i, \tilde{u}),$$
$$\bar{R}_i(b_i, \tilde{a}_i, \tilde{u}) = \underset{b_{-i}, \tilde{a}_{-i}}{\mathrm{E}} \left[ R_i(b, \tilde{a}, \tilde{u}) \right] = \bar{f}_i^{(K+1)}(b_i, \tilde{a}_i, \tilde{u}).$$

We abuse notation and employ $\bar{M}$ to denote the set of ads selected using metric $\bar{R}_i(b_i, \tilde{a}_i, \tilde{u})$.

## 4.2 Revenue Approximation

Next, we derive approximation results for the revenue ranking index. Based on the approximation results, we can calculate the size of $\bar{M}$ needed to guarantee a certain fraction of the optimal revenue.

### 4.2.1 Uniform Distribution.
We start with the case where $\bar{f}_i^{(K+1)}$ and $\bar{f}_i^{(1)}$ are i.i.d random variables that follow the uniform distribution over intervals $[a^{(K+1)}, b^{(K+1)}]$ and $[a^{(1)}, b^{(1)}]$ respectively, for all ads.

LEMMA 6. *Suppose $\bar{f}_i^{(K+1)}$ and $\bar{f}_i^{(1)}$ are i.i.d random variables that follow uniform distributions $U[a^{(K+1)}, b^{(K+1)}]$ and $U[a^{(1)}, b^{(1)}]$. Then always including the ads with top $m$ ranking indices achieves an $\alpha$ fraction of the optimal revenue if:*

$$m \geq \frac{b^{(K+1)} - b^{(1)}}{2(b^{(K+1)} - a^{(K+1)} - b^{(1)} + a^{(1)})} \left[ 2n + 2 - \sqrt{1 - \alpha}(2n + 1) \right].$$

### 4.2.2 General Distribution.
In addition to uniform distributions, we also give results for general distributions under the assumption that $\bar{f}_i^{(K+1)}$ and $\bar{f}_i^{(1)}$ are i.i.d. random variables.

LEMMA 7. *Suppose $\bar{f}_i^{(K+1)}$ and $\bar{f}_i^{(1)}$ are random variables that follow distributions with means $\mu_{(K+1)}, \mu_{(1)}$ and variances $\sigma_{(K+1)}^2, \sigma_{(1)}^2$. Then always including the ads with top $m$ ranking indices $\bar{r}_i$ achieves an $\alpha$ fraction of the optimal revenue if:*

$$m > \alpha n - \frac{\sigma_{(K+1)}}{2\mu_{(K+1)}} n + \frac{\sqrt{2}}{4}(2n + 1)\sqrt{\frac{\sigma_{(K+1)}}{\mu_{(K+1)}}} + \sqrt{\frac{(n\alpha + 1)(1 - \alpha)\mu_{(1)}}{\mu_{(K+1)}}}.$$

## 5 Learning Based Selection Metrics

In previous sections, we proposed the ad selection metric based on their contribution to the objectives. In practice, we use a neural network to approximate the actual contribution of each advertiser, denoted by $\bar{f}^\theta(b_i, \tilde{a}_i, \tilde{u}_i)$. We use supervised learning to update the model's parameter $\theta$. For each auction sample, the input of the learning model includes the advertiser's bid $b_i$, the partial ad features $\tilde{a}_i$, and partial user $\tilde{u}$. The label is $y_i = b_i \times q_i \times \mathbb{I} \left\{ i \in Top_N^K(b, q) \right\}$ or $y_i = b_i \times q_i \times \mathbb{I} \left\{ i \in Top_N^{K+1}(b, q) \right\}$, depending on the objective is welfare or revenue. Note that during the training process, we can obtain the accurate CTR $q_i$ of each ad from the refined CTR model $\mathcal{M}^r$. Thus for any auction sample, we can also determine whether an advertiser is in the top $K$.

Given a set of auction samples $\mathcal{D}_f$, we minimize the mean square error (MSE) between the prediction of $\bar{f}^\theta$ and the label $y_i$, so the loss function is:

$$\mathcal{L} = \frac{1}{|\mathcal{D}_f|} \sum_{j \in \mathcal{D}_f} \sum_{i \in N} \left( \bar{f}^\theta(b_i^j, \tilde{a}_i^j, \tilde{u}_i^j) - y_i^j \right)^2,$$

where the superscript $j$ means the $j$-th auction sample in $\mathcal{D}_f$.

## 6 Experiments

In this section, we conduct extensive experiments on both synthetic and industrial data to evaluate the effectiveness of our proposed selection metrics Max-Wel and Max-Rev.

*Synthetic Data.* We generate synthetic auction data based on the iPinYou [18] dataset, the only publicly available dataset on display advertising released by a major demand-side platform. This dataset comprises logs of bidding, impressions, clicks, and final conversions from 3 campaign seasons, including 78 million bid records and 24 million impression records. As the data from the

**Table 1: Experiment results of different methods on synthetic data.** $n = 5, m = 4$.

| Method | Wel@1 | Wel@2 | Wel@3 | Method | Rev@1 | Rev@2 | Rev@3 |
|---|---|---|---|---|---|---|---|
| REG-CTR | 0.9661 | 0.9534 | 0.9372 | REG-CTR | 0.9301 | 0.9056 | 0.8715 |
| REG | 0.9972 | 0.9907 | 0.9752 | REG | 0.9827 | 0.9543 | 0.9157 |
| PAS | 0.9997 | 0.9995 | 0.9909 | PAS | 0.9389 | 0.9568 | 0.9226 |
| Max-Wel (ours) | **0.9998** | **0.9995** | **0.9984** | Max-Rev | **0.9980** | **0.9970** | **0.9851** |

**Table 2: Experiment results of different methods on industrial data.** $n = 400, m = 10$.

| Method | Wel@1 | Wel@3 | Wel@5 | Method | Rev@1 | Rev@3 | Rev@5 |
|---|---|---|---|---|---|---|---|
| REG-CTR | 0.9434 | 0.8684 | 0.8298 | REG-CTR | 0.8400 | 0.8076 | 0.7627 |
| REG | 0.9661 | 0.8740 | 0.8343 | REG | 0.8236 | 0.7938 | 0.7703 |
| PAS | 0.9161 | 0.8810 | 0.8346 | PAS | 0.8747 | 0.8186 | 0.7817 |
| Max-Wel (ours) | **0.9720** | **0.9018** | **0.8738** | Max-Rev | **0.9217** | **0.8431** | **0.8028** |

first season lacks Advertiser ID and user profile information, and the bidding log lacks paying price data, we select one day's impression log data from the second season to conduct our experiments. The data includes 5 distinct Advertiser IDs (5 bidders), 1.6 million users, and 1.8 million bid records. The full feature of a user $u$ encompasses iPinYou ID, Region ID, City ID, and User Profile ID, while the full ad feature $a_i$ includes Advertiser ID and Creative ID. We assume that the partial user feature $\tilde{u}$ comprises iPinYou ID and Region ID, whereas the partial ad feature $\tilde{a}_i$ includes only Advertiser ID. As the bidding prices were scaled before the release, we regard the paying price as their bids and fit a log-normal distribution to them, thereby simulating the advertisers' bidding strategy. Based on these data, we generate 100,000 auction instances, each comprising a randomly selected user and 5 advertisers. Each advertiser's bid is independently drawn from the fitted log-normal distribution. To ensure alignment between the highest bidding advertiser and the original impression winner in the data, we swap the highest bid within a sampled bid vector with the bid of the winner.

To obtain the allocation outcome of these instances, we need to simulate the GSP auction in the second stage. Before that, we must first train a refined CTR estimator $\mathcal{M}^r$ to generate $q_i$ using the full features of the ad $a_i$ and user $u$. Then, we use $b_i \times q_i$ as the ranking score in the second stage GSP auction. We put detailed descriptions of the training data for the CTR model in Appendix B.1.

*Industrial Data.* The industrial data is sourced from the ad auction log within a major auction platform. We extract a sample of 80,000 ad requests from the logged data in April 2024. In each ad request from a user, about 400 ads compete for exposure. The features for each ad include: 1) attributes specific to the ad itself, such as bid price $b_i$, task type, corporation type, etc.; 2) cross features of the ad and user, like the click-through rate (CTR), the conversion rate (CVR). We regard CTR as $q_i$ and use it to generate the label $b_i \times q_i$ for each ad. As cross features may encompass information from the full features, we opt to only consider attribute features when selecting ad features.

*Evaluation.* We evaluate the performance of different two-stage methods from the perspectives of welfare and revenue respectively.

- Welfare rate: $\text{Wel}@K = \sum_{i=1}^{K} s_M^{(i)} / \sum_{j=1}^{K} s_N^{(j)}$.
- Revenue rate: $\text{Rev}@K = \sum_{i=1}^{K} s_M^{(i+1)} / \sum_{j=1}^{K} s_N^{(j+1)}$.

Recall that $s_M^{(i)}$ represents the $i$-th highest score in the selected ad set $M$, while $s_N^{(j)}$ denotes the $j$-th highest score in the entire set $N$. Note that revenue is computed under the GSP auction, hence the revenue of a specific set $T$ can be expressed as $REV(T) = \sum_{k=1}^{K} s_T^{(k+1)}$.

*Baseline Methods.* To show the effectiveness of our proposed Max-Wel and Max-Rev, we introduce the following two-stage methods as baselines.

- REG-CTR, which trains a regression model with $\tilde{a}_i, \tilde{u}$ as input, $q_i$ as the label and the rank score is bid times the output of the regression model.
- REG, which trains a regression model with $b_i, \tilde{a}_i, \tilde{u}$ as input, $b_i \times q_i$ as the label, and the rank score is the output of the model.
- PAS [26], which uses $b_i, \tilde{a}_i, \tilde{u}$ as input and outputs the probability of each ad being in TopK.

All these baselines are also restricted to use the same partial features $\langle \tilde{a}_i, \tilde{u} \rangle$. The neural network architecture remains consistent across all methods, with nearly identical input. The only distinction lies in the REG-CTR network, which lacks the bid input.

*Performance Comparison.* Results of different methods on synthetic data and industrial data are given in Table 1 and 2. All the results are obtained by averaging across 10 runs with distinctive seeds. We omit to record the standard deviation as it consistently remains below 1% across various evaluation metrics for all methods in our experiments.

It is evident from Table 1 and 2 that our method outperforms all baseline methods in both data settings, and in terms of both welfare and revenue. For instance, in the experiments with industrial data, our Max-Wel improves the welfare rate by 0.59%, 2.08%, 3.92% compared to the best performance among other baseline methods for $K = 1, 3, 5$, respectively. The reasons why our methods outperform baseline methods are two-fold: 1) compared to PAS, which predicts the probability of each ad being among the top $K$, our method takes


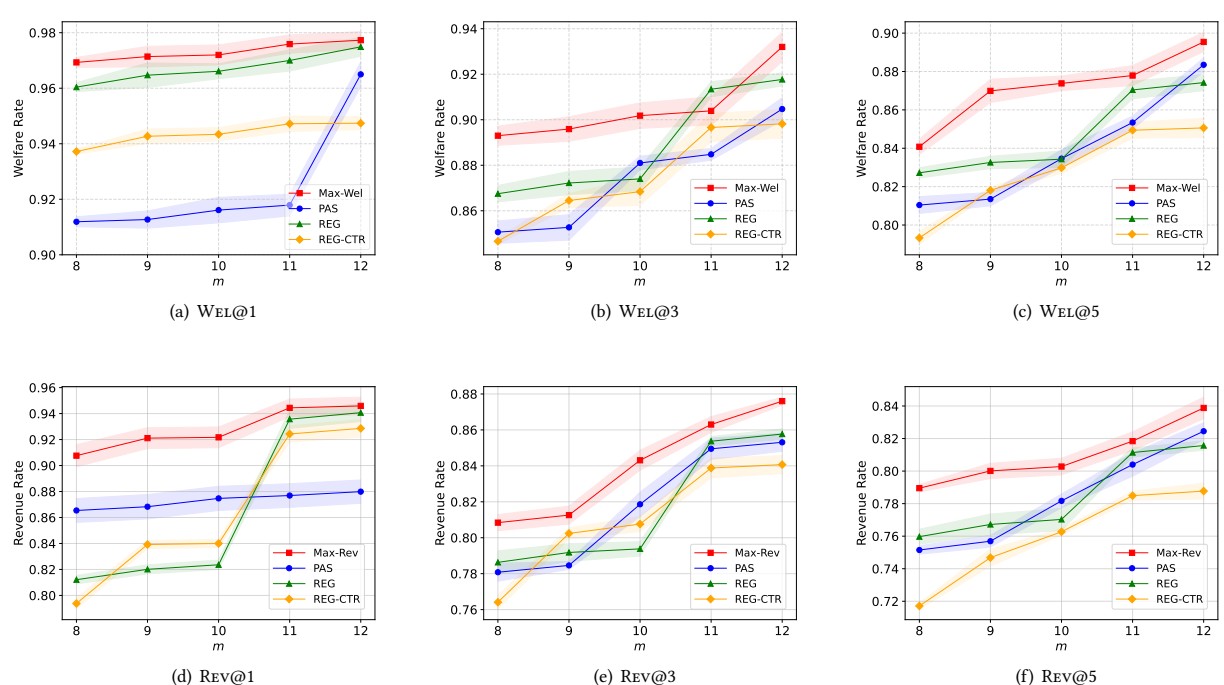

**Figure 1: Experiment results of different methods on industrial data with different $m$.**

**Table 3: Violation rate of perturbation test, with unit $\times 10^{-4}$.**

|  | REG-CTR | REG | PAS | Max-Wel | Max-Rev |
|---|---|---|---|---|---|
| Violation Rate | 0 | $11.04 \pm 4.04$ | $8.12 \pm 3.18$ | $3.31 \pm 2.36$ | $4.78 \pm 1.87$ |

into account not only whether each ad is in the top $K$, but also how much each ad contributes to the objective; 2) compared to REG-CTR and REG, our methods focus on advertisers who contribute more to the objective, hence can more accurately screen out high-quality advertisers.

Moreover, we compare the performance of different methods with different $m$ under the industrial data, and results are shown in Figure 1. The experimental results demonstrate the superiority of our proposed method over other baselines in terms of both welfare and revenue metrics across various values of $m$. Particularly noteworthy is the observation that the margin of improvement tends to be greater with smaller $m$. This indicates that our methods excel in prioritizing high-quality advertisers, further validating their effectiveness in selecting top-performing advertisers.

*IC Testing.* The IC property requires the allocation for each ad $i$ to be monotone increasing to bid $b_i$. To test to what extent different methods satisfy the IC condition, we employ the commonly used IC test in ad auctions [6, 7] by perturbating each advertiser's bid and evaluating the violation rate. Specifically, for each ad, all features remain the same, except that $b_i$ is replaced by $b_i \times \alpha$, where $\alpha \in \mathcal{S}_p = \{0.2x \mid x = 1, \ldots, 10\}$ is a perturbation factor. All features of other ads remain unchanged. A test does not violate IC test if

$\exists \alpha_0 \in \mathcal{S}_p$ such that ad $i$ can enter the second stage with $b_i \times \alpha$ for all $\alpha \geq \alpha_0$; or if ad $i$ cannot enter the second stage for any $\alpha \in \mathcal{S}_p$.

We sample 1000 auctions from the test set and conduct the IC test on each ad in each auction for all methods. The results are shown in Table 3. The results indicate that our methods exhibit low violation rates, which means that even without using specialized structures to ensure the monotonicity of the learning model, our proposed metrics guarantee approximate monotonicity. Notably, the REG-CTR method ranks ads by multiplying the bid with the learned model's output, inherently preserving monotonicity.

## 7 Conclusion

We study the design of two-stage auctions from the angle of optimizing welfare and revenue respectively. We explicitly derive each ad's contribution to each objective function, and use this as a selection metric for bidder selection in the first stage. We provide theoretical guarantees for our metrics under both uniform and general distributions. We demonstrate that these metrics can be effectively learned using neural networks. Experimental results on both synthetic and industrial data show that our methods significantly outperform existing approaches, highlighting the advantages of selecting top-performing advertisers.

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

# Appendix

## A  Omitted Proofs

### A.1  Proof of Lemma 1

Proof. It suffices to show that:

$$\text{WEL}(M|\boldsymbol{b}, \tilde{\boldsymbol{a}}, \tilde{u}) = \sum_{i \in M} \mathop{\mathbf{E}}_{\boldsymbol{q}|\tilde{\boldsymbol{q}}} \left[ b_i q_i \mathbb{I} \left\{ i \in Top_M^K(\boldsymbol{b}, \boldsymbol{q}) \right\} \right]$$

$$\geq \sum_{i \in M} \mathop{\mathbf{E}}_{\boldsymbol{q}|\tilde{\boldsymbol{q}}} \left[ b_i q_i \mathbb{I} \left\{ i \in Top_N^K(\boldsymbol{b}, \boldsymbol{q}) \right\} \right]$$

$$= \sum_{i \in M} f_i(\boldsymbol{b}, \tilde{\boldsymbol{a}}, \tilde{u}).$$

The inequality comes from that for any ad $i$ in the selected set $M$, (1) if $i$ is originally in $Top_N^K(\boldsymbol{b}, \boldsymbol{q})$, then it must also in the set $Top_M^K(\boldsymbol{b}, \boldsymbol{q})$; (2) if $i$ is not in $Top_N^K(\boldsymbol{b}, \boldsymbol{q})$, then $\mathbb{I}\left\{i \in Top_N^K(\boldsymbol{b}, \boldsymbol{q})\right\}$ must equal to 0, but it may still be in the $Top_M^K(\boldsymbol{b}, \boldsymbol{q})$. Therefore, the expected welfare contribution of set $M$ is a lower bound of the actual welfare of choosing set $M$. □

### A.2  Proof of Lemma 2

Proof. Let $\bar{f}_{(i)}$ be the $i$-th order statistic (i.e., the $i$-th smallest value) of $\{\bar{f}_i\}_{i=1}^n$.

It is known that if a random variable $X_i$ follows $U[0, 1]$, then the $j$-th order statistic of $n$ independent samples $\{X_i\}_{i=1}^n$ follows a Beta distribution $\text{Beta}(j, n-j+1)$ with mean $\frac{j}{n+1}$. Each $\bar{f}_j = (b-a)X_j + a$ can be viewed as an affine transformation of $X_j$. So the expectation of $\bar{f}_{(j)}$ is:

$$\mathbf{E}\left[\bar{f}_{(j)}\right] = (b-a)\frac{j}{n+1} + a.$$

If we include the top $m$ ads in $\bar{M}$, we have:

$$\mathbf{E}\left[\sum_{j=n-m+1}^n \bar{f}_{(j)}\right] = \sum_{j=n-m+1}^n \mathbf{E}\left[\bar{f}_{(j)}\right]$$

$$= (b-a)\frac{m(2n-m+1)}{2(n+1)} + ma. \quad (7)$$

If we are allowed to include all ads in the first stage, we can still obtain the optimal social welfare by setting $M = N$:

$$\mathbf{E}\left[\sum_{j=1}^n \bar{f}_{(j)}\right] = \mathbf{E}\left[\sum_{j=1}^n \bar{f}_j\right]$$

$$= \sum_{j=1}^n \mathbf{E}\left[\bar{f}_j\right]$$

$$= \frac{n(b+a)}{2}. \quad (8)$$

Therefore, to guarantee an $\alpha$ fraction of the optimal welfare, we need to ensure that:

$$\mathbf{E}\left[\sum_{j=n-m+1}^n \bar{f}_{(j)}\right] \geq \alpha\,\mathbf{E}\left[\sum_{j=1}^n \bar{f}_{(j)}\right],$$

which is equivalent to:

$$-(b-a)m^2 + \eta m - \zeta \geq 0, \quad (9)$$

where $\eta = (b-a)(2n+1) + a(2n+2)$ and $\zeta = \alpha n(n+1)(b+a)$. Solving the quadratic inequality (9), we obtain:

$$m \geq \frac{\eta - \sqrt{\eta^2 - 4(b-a)\zeta}}{2(b-a)}.$$

Now it suffices to show that the above inequality can be implied by inequality (2), i.e.,

$$\eta - \sqrt{\eta^2 - 4(b-a)\zeta} \leq b\left[2n+2 - \sqrt{1-\alpha}(2n+1)\right].$$

To prove the above inequality, note that:

$$\eta < (b-a)(2n+2) + a(2n+2) = b(2n+2)$$
$$\eta > (b-a)(2n+1) + a(2n+1) = b(2n+1),$$

and

$$4(b-a)\zeta = 4\alpha(b-a)(b+a)n(n+1) < \alpha b^2(2n+1)^2.$$

Therefore,

$$\eta - \sqrt{\eta^2 - 4(b-a)\zeta} \leq b(2n+2) - \sqrt{b^2(2n+1)^2 - \alpha b^2(2n+1)^2}$$

$$= b(2n+2) - b(2n+1)\sqrt{1-\alpha}$$

$$= b\left[2n+2 - \sqrt{1-\alpha}(2n+1)\right].$$

□

### A.3  Proof of Lemma 3

The proof of Lemma 3 makes use of the following result:

Lemma 8 ([1, 3]). *Let $\{X_i\}_{i=1}^n$ be $n$ i.i.d. random variables each with mean $\mu$ and variance $\sigma^2$. The $j$-th order statistic $X_{(j)}$ satisfies:*

$$\mathbf{E}\left[X_{(j)}\right] \leq \mu + \sigma\sqrt{\frac{j-1}{n-j+1}}. \quad (10)$$

Proof of Lemma 3. We consider the welfare loss for only including the top $m$ ads. According to Lemma 8, the loss can be bounded as:

$$\mathbf{E}\left[\sum_{j=1}^{n-m} \bar{f}_{(j)}\right] = \sum_{j=1}^{n-m} \mathbf{E}\left[\bar{f}_{(j)}\right]$$

$$\leq (n-m)\mu + \sigma\sum_{j=1}^{n-m}\sqrt{\frac{j-1}{n-j+1}}.$$

Using the Taylor expansion of $\sqrt{x}$ at $x = 1$, one can easily verify that $\sqrt{x} \leq \frac{x+1}{2}$ for all $x \geq 0$. Plugging into the above equation gives:

$$\mathbf{E}\left[\sum_{j=1}^{n-m} \bar{f}_{(j)}\right] \leq (n-m)\mu + \frac{\sigma n}{2}\sum_{j=1}^{n-m}\frac{1}{n-j+1}$$

$$\leq (n-m)\mu + \frac{\sigma n}{2}\sum_{j=1}^{n-m}\frac{1}{m+1}$$

$$= (n-m)\mu + \frac{\sigma n(n-m)}{2(m+1)}.$$

Similarly, if we are allowed to include all ads in the first stage, we can achieve the optimal welfare, which is:

$$\mathbf{E}\left[\sum_{j=1}^{n} \bar{f}_{(j)}\right] = \mathbf{E}\left[\sum_{j=1}^{n} \bar{f}_j\right]$$
$$= \sum_{j=1}^{n} \mathbf{E}\left[\bar{f}_j\right]$$
$$= n\mu.$$

To achieve an $\alpha$ fraction of the optimal welfare, we need to ensure that the loss is no more than $1 - \alpha$ fraction of the optimal welfare, i.e.,

$$(n - m)\mu + \frac{\sigma n(n - m)}{2(m + 1)} \le (1 - \alpha)n\mu,$$

or equivalently,

$$-2\mu m^2 + \eta m + \zeta \le 0,$$

where $\eta = 2\alpha n\mu - 2\mu - \sigma n$ and $\zeta = \sigma n^2 + 2\alpha n\mu$. The solution to the quadratic inequality is:

$$m \ge \frac{\eta + \sqrt{\eta^2 + 8\mu\zeta}}{4\mu}.$$

To prove the lemma, we need to show that inequality (3) implies the above inequality, i.e.,

$$\frac{\eta + \sqrt{\eta^2 + 8\mu\zeta}}{4\mu} < \alpha n - \frac{\sigma}{2\mu}n + \frac{\sqrt{2}}{4}(2n + 1)\sqrt{\frac{\sigma}{\mu}}.$$

Notice that

$$\eta^2 + 8\mu\zeta = (2\alpha n\mu - 2\mu - \sigma n)^2 + 8\mu(\sigma n^2 + 2\alpha n\mu)$$
$$= [(2\alpha\mu - \sigma)n + 2\mu]^2 + 8\mu\sigma n^2 + 8\mu\sigma n$$
$$< [(2\alpha\mu - \sigma)n + 2\mu]^2 + 8\mu\sigma\left(n + \frac{1}{2}\right)^2$$
$$< \left[(2\alpha\mu - \sigma)n + 2\mu + \sqrt{8\mu\sigma}\left(n + \frac{1}{2}\right)\right]^2.$$

Therefore,

$$\frac{\eta + \sqrt{\eta^2 + 8\mu\zeta}}{4\mu} < \frac{\eta + (2\alpha\mu - \sigma)n + 2\mu + \sqrt{8\mu\sigma}\left(n + \frac{1}{2}\right)}{4\mu}$$
$$= \alpha n - \frac{\sigma}{2\mu}n + \frac{\sqrt{2}}{4}(2n + 1)\sqrt{\frac{\sigma}{\mu}}.$$

□

## A.4 Proof of Lemma 5

PROOF. It suffices to show:

$$\text{Rev}(M|\boldsymbol{b}, \tilde{\boldsymbol{a}}, \tilde{\boldsymbol{u}})$$
$$= \sum_{i \in M} \mathop{\mathbf{E}}_{\boldsymbol{q}|\tilde{\boldsymbol{q}}}\left[s_i \mathbb{I}\left\{i \in Top_M^{K+1}(\boldsymbol{b}, \boldsymbol{q})\right\}\right] - \sum_{i \in M} \mathop{\mathbf{E}}_{\boldsymbol{q}|\tilde{\boldsymbol{q}}}\left[s_i \mathbb{I}\left\{i \in Top_M^1(\boldsymbol{b}, \boldsymbol{q})\right\}\right]$$
$$\ge \sum_{i \in M} \mathop{\mathbf{E}}_{\boldsymbol{q}|\tilde{\boldsymbol{q}}}\left[s_i \mathbb{I}\left\{i \in Top_N^{K+1}(\boldsymbol{b}, \boldsymbol{q})\right\}\right] - \sum_{i \in M} \mathop{\mathbf{E}}_{\boldsymbol{q}|\tilde{\boldsymbol{q}}}\left[s_i \mathbb{I}\left\{i \in Top_N^1(\boldsymbol{b}, \boldsymbol{q})\right\}\right]$$
$$= \sum_{i \in M} r_i(\boldsymbol{b}, \tilde{\boldsymbol{a}}, \tilde{\boldsymbol{u}}).$$

Note that the actual revenue is the sum of scores from $s_M^{(2)}$ to $s_M^{(K+1)}$. Thus we have:

$$\sum_{i \in M} \mathop{\mathbf{E}}_{\boldsymbol{q}|\tilde{\boldsymbol{q}}}\left[s_i \mathbb{I}\left\{i \in Top_M^{K+1}(\boldsymbol{b}, \boldsymbol{q})\right\}\right] - \sum_{i \in M} \mathop{\mathbf{E}}_{\boldsymbol{q}|\tilde{\boldsymbol{q}}}\left[s_i \mathbb{I}\left\{i \in Top_M^1(\boldsymbol{b}, \boldsymbol{q})\right\}\right]$$
$$= \sum_{i \in M} \mathop{\mathbf{E}}_{\boldsymbol{q}|\tilde{\boldsymbol{q}}}\left[s_i \mathbb{I}\left\{i \in Top_M^{K+1}(\boldsymbol{b}, \boldsymbol{q})\right\} - s_i \mathbb{I}\left\{i \in Top_M^1(\boldsymbol{b}, \boldsymbol{q})\right\}\right]$$
$$= \sum_{i \in M} \mathop{\mathbf{E}}_{\boldsymbol{q}|\tilde{\boldsymbol{q}}}\left[s_i \mathbb{I}\left\{i \in Top_M^{K+1}(\boldsymbol{b}, \boldsymbol{q}) \& i \notin Top_M^1(\boldsymbol{b}, \boldsymbol{q})\right\}\right].$$

For any ad $i$ in the selected set $M$, (1) if $i$ is originally in $Top_N^{K+1}(\boldsymbol{b}, \boldsymbol{q}) - Top_N^1(\boldsymbol{b}, \boldsymbol{q})$, then it must also in the set $Top_M^{K+1}(\boldsymbol{b}, \boldsymbol{q}) - Top_M^1(\boldsymbol{b}, \boldsymbol{q})$; (2) if $i$ is not in $Top_N^{K+1}(\boldsymbol{b}, \boldsymbol{q}) - Top_N^1(\boldsymbol{b}, \boldsymbol{q})$, it may still be in the set $Top_M^{K+1}(\boldsymbol{b}, \boldsymbol{q}) - Top_M^1(\boldsymbol{b}, \boldsymbol{q})$. Then we have:

$$\sum_{i \in M} \mathop{\mathbf{E}}_{\boldsymbol{q}|\tilde{\boldsymbol{q}}}\left[s_i \mathbb{I}\left\{i \in Top_M^{K+1}(\boldsymbol{b}, \boldsymbol{q}) \& i \notin Top_M^1(\boldsymbol{b}, \boldsymbol{q})\right\}\right]$$
$$\ge \sum_{i \in M} \mathop{\mathbf{E}}_{\boldsymbol{q}|\tilde{\boldsymbol{q}}}\left[s_i \mathbb{I}\left\{i \in Top_N^{K+1}(\boldsymbol{b}, \boldsymbol{q}) \& i \notin Top_N^1(\boldsymbol{b}, \boldsymbol{q})\right\}\right]$$
$$= \sum_{i \in M} \mathop{\mathbf{E}}_{\boldsymbol{q}|\tilde{\boldsymbol{q}}}\left[s_i \mathbb{I}\left\{i \in Top_N^{K+1}(\boldsymbol{b}, \boldsymbol{q})\right\}\right] - \sum_{i \in M} \mathop{\mathbf{E}}_{\boldsymbol{q}|\tilde{\boldsymbol{q}}}\left[s_i \mathbb{I}\left\{i \in Top_N^1(\boldsymbol{b}, \boldsymbol{q})\right\}\right]$$
$$= \sum_{i \in M} r_i(\boldsymbol{b}, \tilde{\boldsymbol{a}}, \tilde{\boldsymbol{u}}).$$

Then we complete our proof.              □

## A.5 Proof of Lemma 6

PROOF. Let $\bar{f}_{(i)}^{(K+1)}$ be the $i$-th order statistic, that is, the $i$-th smallest value of $\{\bar{f}_i^{(K+1)}\}_{i=1}^{n}$, and $\bar{f}_{(i)}^{(1)}$ be the $i$-th order statistic of $\{\bar{f}_i^{(1)}\}_{i=1}^{n}$.

It is known that if a random variable $X_i$ follows $U[0, 1]$, then the $j$-th order statistic of $n$ independent samples $\{X_i\}_{i=1}^{n}$ follows a Beta distribution $\text{Beta}(j, n - j + 1)$ with mean $\frac{j}{n+1}$. Each $\bar{f}_j^{(K+1)} = (b^{(K+1)} - a^{(K+1)})X_j + a^{(K+1)}$ can be viewed as an affine transformation of $X_j$. So the expectation of $\bar{f}_{(j)}^{(K+1)}$ is:

$$\mathbf{E}\left[\bar{f}_{(j)}^{(K+1)}\right] = (b^{(K+1)} - a^{(K+1)})\frac{j}{n + 1} + a^{(K+1)}.$$

Similarly, the expectation of $\bar{f}_{(j)}^{(1)}$ is:

$$\mathbf{E}\left[\bar{f}_{(j)}^{(1)}\right] = (b^{(1)} - a^{(1)})\frac{j}{n + 1} + a^{(1)}.$$

If we include top $m$ ads in $\bar{M}$, we have:

$$\mathbf{E}\left[\sum_{j=n-m+1}^{n} \bar{r}_{(j)}\right]$$

$$= \mathbf{E}\left[\sum_{j=n-m+1}^{n} \bar{f}_{(j)}^{(K+1)} - \bar{f}_{(j)}^{(1)}\right]$$

$$= \sum_{j=n-m+1}^{n} \mathbf{E}\left[\bar{f}_{(j)}^{(K+1)}\right] - \sum_{j=n-m+1}^{n} \mathbf{E}\left[\bar{f}_{(j)}^{(1)}\right]$$

$$= (b^{(K+1)} - a^{(K+1)} - b^{(1)} + a^{(1)}) \frac{m(2n-m+1)}{2(n+1)} + m(a^{(K+1)} - a^{(1)}). \tag{11}$$

If we are allowed to include all ads in the first stage, we can obtain the optimal revenue by setting $M = N$:

$$\mathbf{E}\left[\sum_{j=1}^{n} \bar{r}_{(j)}\right] = \mathbf{E}\left[\sum_{j=1}^{n} \bar{f}_{(j)}^{(K+1)} - \bar{f}_{(j)}^{(1)}\right]$$

$$= \sum_{j=1}^{n} \mathbf{E}\left[\bar{f}_{(j)}^{(K+1)}\right] - \sum_{j=1}^{n} \mathbf{E}\left[\bar{f}_{(j)}^{(1)}\right]$$

$$= \frac{n(b^{(K+1)} + a^{(K+1)} - b^{(1)} - a^{(1)})}{2}. \tag{12}$$

Therefore, to guarantee an $\alpha$ fraction of the optimal welfare, we need to ensure that:

$$\mathbf{E}\left[\sum_{j=n-m+1}^{n} \bar{r}_{(j)}\right] \geq \alpha \mathbf{E}\left[\sum_{j=1}^{n} \bar{r}_{(j)}\right],$$

which is equivalent to:

$$-(b^{(K+1)} - a^{(K+1)} - b^{(1)} + a^{(1)})m^2 + \eta m - \zeta \geq 0, \tag{13}$$

where $\eta = (b^{(K+1)} - a^{(K+1)} - b^{(1)} + a^{(1)})(2n+1) + (a^{(K+1)} - a^{(1)})(2n+2)$ and $\zeta = \alpha n(n+1)(b^{(K+1)} + a^{(K+1)} - b^{(1)} - a^{(1)})$. Solving the quadratic inequality (13), we obtain:

$$m \geq \frac{\eta - \sqrt{\eta^2 - 4(b^{(K+1)} - a^{(K+1)} - b^{(1)} + a^{(1)})\zeta}}{2(b^{(K+1)} - a^{(K+1)} - b^{(1)} + a^{(1)})}.$$

Note that:

$$\eta < (b^{(K+1)} - a^{(K+1)} - b^{(1)} + a^{(1)})(2n+2) + (a^{(K+1)} - a^{(1)})(2n+2)$$

$$= (b^{(K+1)} - b^{(1)})(2n+2)$$

$$\eta > (b^{(K+1)} - a^{(K+1)} - b^{(1)} + a^{(1)})(2n+1) + (a^{(K+1)} - a^{(1)})(2n+1)$$

$$= (b^{(K+1)} - b^{(1)})(2n+1),$$

and

$$4(b^{(K+1)} - a^{(K+1)} - b^{(1)} + a^{(1)})\zeta$$

$$= 4\alpha\left[b^{(K+1)} - a^{(K+1)} - (a^{(K+1)} - a^{(1)})\right] \cdot \left[b^{(K+1)} - a^{(K+1)}\right.$$

$$\left. + (a^{(K+1)} - a^{(1)})\right] \cdot n(n+1)$$

$$< \alpha\left[b^{(K+1)} - b^{(1)}\right]^2 \cdot (2n+1)^2.$$

Therefore,

$$\eta - \sqrt{\eta^2 - 4(b^{(K+1)} - a^{(K+1)} - b^{(1)} + a^{(1)})\zeta}$$

$$\leq (b^{(K+1)} - b^{(1)})(2n+2) - (b^{(K+1)} - b^{(1)})(2n+1)\sqrt{1-\alpha}$$

$$= (b^{(K+1)} - b^{(1)})\left[2n+2 - \sqrt{1-\alpha}(2n+1)\right].$$

Then we obtain:

$$m \geq \frac{b^{(K+1)} - b^{(1)}}{2(b^{(K+1)} - a^{(K+1)} - b^{(1)} + a^{(1)})}\left[2n+2 - \sqrt{1-\alpha}(2n+1)\right],$$

which proves the lemma. $\qquad\square$

## A.6 Proof of Lemma 7

PROOF. The revenue upper bound can be achieved by allowing to include all ads in the first stage, that is:

$$\mathbf{E}\left[\sum_{j=1}^{n} \bar{r}_{(j)}\right] = \sum_{j=1}^{n} \mathbf{E}\left[\bar{f}_{(j)}^{(K+1)}\right] - \sum_{j=1}^{n} \mathbf{E}\left[\bar{f}_{(j)}^{(1)}\right]$$

$$= \sum_{j=1}^{n} \mathbf{E}\left[\bar{f}_{j}^{(K+1)}\right] - \sum_{j=1}^{n} \mathbf{E}\left[\bar{f}_{j}^{(1)}\right]$$

$$= n(\mu_{(K+1)} - \mu_{(1)}).$$

To achieve an $\alpha$ fraction of the optimal revenue, we need to ensure that the actual revenue of selecting set $M$ is greater than $\alpha$ fraction of the optimal revenue, that is:

$$\text{REV}(M|\boldsymbol{b}, \tilde{\boldsymbol{a}}, \tilde{u}) \geq \alpha\left(\sum_{j=1}^{n} \mathbf{E}\left[\bar{f}_{j}^{(K+1)}\right] - \sum_{j=1}^{n} \mathbf{E}\left[\bar{f}_{j}^{(1)}\right]\right),$$

or equivalently, we ensure the revenue loss is less than $1-\alpha$ fraction of the optimal revenue.

According to Lemma 5, we have:

$$\text{REV}(M|\boldsymbol{b}, \tilde{\boldsymbol{a}}, \tilde{u}) \geq \sum_{i \in M} r_i(\boldsymbol{b}, \tilde{\boldsymbol{a}}, \tilde{u})$$

$$= \sum_{i \in M} f_i^{(K+1)} - \sum_{i \in M} f_i^{(1)}$$

$$\geq \sum_{i \in M} f_i^{(K+1)} - \sum_{i \in N} f_i^{(1)}.$$

Then the revenue loss is bounded by:

$$\sum_{i \in N-M} r_i(\boldsymbol{b}, \tilde{\boldsymbol{a}}, \tilde{u}) = \sum_{i \in N-M} f_i^{(K+1)} - \sum_{i \in N-M} f_i^{(1)}$$

$$\leq \sum_{i \in N-M} f_i^{(K+1)}$$

$$= \mathbf{E}\left[\sum_{j=1}^{n-m} f_{(j)}^{(K+1)}\right].$$

Therefore, it suffices to show that:

$$\mathbf{E}\left[\sum_{j=1}^{n-m} f_{(j)}^{(K+1)}\right] \leq (1-\alpha)\left(\sum_{j=1}^{n} \mathbf{E}\left[\bar{f}_{j}^{(K+1)}\right] - \sum_{j=1}^{n} \mathbf{E}\left[\bar{f}_{j}^{(1)}\right]\right).$$

According to Lemma 8, we have:

$$\mathbf{E}\left[\sum_{j=1}^{n-m} f_{(j)}^{(K+1)}\right] = \sum_{j=1}^{n-m} \mathbf{E}\left[f_{(j)}^{(K+1)}\right]$$

$$\leq (n-m)\mu_{(K+1)} + \sigma_{(K+1)}\sum_{j=1}^{n-m}\sqrt{\frac{j-1}{n-j+1}}.$$

Using the Taylor expansion of $\sqrt{x}$ at $x = 1$, we have $\sqrt{x} \leq \frac{x+1}{2}$ for all $x \geq 0$. Plugging into the above equation gives:

$$\mathbf{E}\left[\sum_{j=1}^{n-m} f_{(j)}^{(K+1)}\right] \leq (n-m)\mu_{(K+1)} + \frac{n\sigma_{(K+1)}}{2}\sum_{j=1}^{n-m}\frac{1}{n-j+1}$$

$$\leq (n-m)\mu_{(K+1)} + \frac{n\sigma_{(K+1)}}{2}\sum_{j=1}^{n-m}\frac{1}{m+1}$$

$$= (n-m)\mu_{(K+1)} + \frac{n\sigma_{(K+1)}(n-m)}{2(m+1)}$$

Then we need to ensure:

$$(n-m)\mu_{(K+1)} + \frac{n\sigma_{(K+1)}(n-m)}{2(m+1)} \leq (1-\alpha)n(\mu_{(K+1)} - \mu_{(1)}),$$

or equivalently,

$$-2\mu_{(K+1)}m^2 + \eta m + \chi \leq 0.$$

where $\eta = 2\alpha n\mu_{(K+1)} - 2\mu_{(K+1)} - \sigma_{(K+1)}n + 2n(1-\alpha)\mu_{(1)}$ and $\chi = \sigma_{(K+1)}n^2 + 2\alpha\mu_{(K+1)}n + 2n(1-\alpha)\mu_{(1)}$. The solution to the quadratic equation is:

$$m \geq \frac{\eta + \sqrt{\eta^2 + 8\mu_{(K+1)}\chi}}{4\mu_{(K+1)}}.$$

To prove the lemma, we need to show that inequality (7) implies the above inequality, i.e.,

$$\frac{\eta + \sqrt{\eta^2 + 8\mu_{(K+1)}\chi}}{4\mu_{(K+1)}}$$

$$< \alpha n - \frac{\sigma_{(K+1)}}{2\mu_{(K+1)}}n + \frac{\sqrt{2}}{4}(2n+1)\sqrt{\frac{\sigma_{(K+1)}}{\mu_{(K+1)}}} + \sqrt{\frac{(n\alpha+1)(1-\alpha)\mu_{(1)}}{\mu_{(K+1)}}}.$$

Notice that

$$\eta^2 + 8\mu_{(K+1)}\chi$$

$$= (2\alpha n\mu_{(K+1)} - 2\mu_{(K+1)} - \sigma_{(K+1)}n + 2n(1-\alpha)\mu_{(1)})^2$$

$$\quad + 8\mu_{(K+1)}\left[\sigma_{(K+1)}n^2 + 2\alpha n\mu_{(K+1)} + 2n(1-\alpha)\mu_{(1)}\right]$$

$$= \left[(2\alpha\mu_{(K+1)} - \sigma_{(K+1)})n - (2\mu_{(K+1)} - 2n(1-\alpha)\mu_{(1)})\right]^2$$

$$\quad + 8\mu_{(K+1)}\sigma_{(K+1)}n^2 + 8\mu_{(K+1)}\sigma_{(K+1)}n$$

$$\quad + (8n\alpha\mu_{(K+1)} - 4n\sigma_{(K+1)} + 8\mu_{(K+1)})(1-\alpha)2n\mu_{(1)}$$

$$< \left[(2\alpha\mu_{(K+1)} - \sigma_{(K+1)})n + (2\mu_{(K+1)} - 2n(1-\alpha)\mu_{(1)})\right]^2$$

$$\quad + 8\mu_{(K+1)}\sigma_{(K+1)}(n + \tfrac{1}{2})^2 + 16n\mu_{(K+1)}(n\alpha+1)(1-\alpha)\mu_{(1)}$$

$$< \left[(2\alpha\mu_{(K+1)} - \sigma_{(K+1)})n + (2\mu_{(K+1)} - 2n(1-\alpha)\mu_{(1)})\right.$$

$$\quad \left. + \sqrt{8\mu_{(K+1)}\sigma_{(K+1)}}(n + \tfrac{1}{2}) + \sqrt{16n\mu_{(K+1)}(n\alpha+1)(1-\alpha)\mu_{(1)}}\right].$$

Put the above inequality back, we have

$$\frac{\eta + \sqrt{\eta^2 + 8\mu_{(K+1)}\chi}}{4\mu_{(K+1)}}$$

$$< \alpha n - \frac{\sigma_{(K+1)}}{2\mu_{(K+1)}}n + \frac{\sqrt{2}}{4}(2n+1)\sqrt{\frac{\sigma_{(K+1)}}{\mu_{(K+1)}}} + \sqrt{\frac{(n\alpha+1)(1-\alpha)\mu_{(1)}}{\mu_{(K+1)}}},$$

which proves the lemma. □

# B  Additional Experiment Details

## B.1  Training Data For CTR Model

It's worth noting that directly using the impression log and click log as training data for $\mathcal{M}^r$ isn't feasible due to the click log's limited 1,289 records compared to the impression log's 1.8 million records. To address this imbalance between click and impression data, we partition the impression data into click and non-click data. We treat every impression and click record as coordinates in a high-dimensional space. Then, we measure the distance between each impression point and its nearest click point. Our rationale is that a closer distance should indicate a higher likelihood of the impression being clicked. In essence, we partition the impression data based on the minimum Euclidean distance between each impression point and its nearest click point. The division ratio is set at $1 : 6$.

## B.2  Experimental Parameters

Both synthetic and industrial data are split into training and test sets with an 8:2 ratio. The neural network structure uses a simple multi-layer perception (MLP) structure with ReLU [10] as the activation function, and all methods share this network structure. We use the Adam optimizer [14] to update the parameters of the neural network. We map discrete features into continuous spaces using embedding with an embedding size of 64. All the experiments are run on a Linux machine with NVIDIA GPU cores.

