# OpenReview forum: "Two-stage Auction Design in Online Advertising"
_ACM.org/TheWebConf/2025/Conference — WWW 2025 Poster_

### Official Review · Reviewer_wW9G · 2024-11-06

**Novelty:** 4
**Technical Quality:** 5

**Review:**

This paper investigated a two-stage auction design in an online ad auction. The motivation for the separated stages is the difficulty of computing all the CTRs for a large number of advertisers. In the first stage, a smaller set of features is used to calculate the CTR, while the complete set is used in the second stage. This paper improved from the naive approach, which uses two CTR models for the first and second stages. The paper dealt with both welfare and revenue maximization. The main idea is to exploit each ad's contribution to the welfare or revenue maximization objectives. Experimental results from synthetic and industrial data show a minor improvement (from less than 1% to nearly 4%) in welfare and revenue over the other approaches.
This paper's contribution is moderate due to minor performance improvements.

**Questions:**

Although I am unsure about it, my main concern is the motivation for a two-stage setting: the difficulty of a platform in calculating the CTRs of a large number of advertisers. Could the platform precompute these CTRs before the auction? As this is not constrained by the time limits of a real-time auction, they can use the refined model to acquire very accurate results. Also, it seems absurd that the CTRs must be calculated repeatedly for each auction.

**Reviewer Confidence:**

3: The reviewer is confident but not certain that the evaluation is correct

**Scope:**

4: The work is relevant to the Web and to the track, and is of broad interest to the community

---

### Official Review · Reviewer_sJs2 · 2024-11-15

**Novelty:** 5
**Technical Quality:** 3

**Review:**

This paper addresses the scalability issues faced by modern online advertising systems, where a large number of advertisers participate in each auction. To manage this, two-stage auctions are commonly used, where a fast but coarse model selects a small subset of advertisers in the first stage, and a slower, more refined model determines the winners in the second stage. However, existing two-stage auction mechanisms focus mainly on optimizing welfare and often overlook other important objectives, such as revenue.
The authors propose two new ad-wise selection metrics—Max-Wel and Max-Rev—which are based on an ad's contribution to either welfare or revenue, respectively. The paper provides theoretical guarantees for these metrics and demonstrates that they are applicable to both welfare and revenue optimization tasks. Additionally, the proposed metrics can be easily implemented using neural networks. Extensive experiments on both synthetic and real-world industrial data show that these new selection metrics outperform existing baselines in terms of efficiency and effectiveness.
Pros:
1.The author’s work is comprehensive, considering not only welfare and revenue but also specific cases for both scenarios in practical applications.
2.The background of this paper is intriguing. The two-stage auction mechanism presents an interesting approach to improving ranking mechanisms, especially when considering aspects like social welfare.
Cons:
1.It would be helpful if the author provided a detailed explanation of the first-stage auction mechanism in the form of pseudocode to aid reader understanding.
2.The problem background is not very clear, especially regarding "advertising feature $a_i$." I find it difficult to understand what the feature represents.

**Questions:**

1.What is the relationship between predicting CTR and selecting ads based on welfare and revenue?
2.I find it difficult to understand the meaning of "partial advertising features" here, as well as the subset relationship between partial and full advertising features.
3.In line 344, the paper mentions that the function f can be trained using a neural network. Could you specify the exact training method, or provide relevant references?

**Reviewer Confidence:**

3: The reviewer is confident but not certain that the evaluation is correct

**Scope:**

3: The work is somewhat relevant to the Web and to the track, and is of narrow interest to a sub-community

---

### Official Review · Reviewer_psxR · 2024-11-26

**Novelty:** 6
**Technical Quality:** 6

**Review:**

This paper proposes novel selection metrics for the two-stage auction problem in online advertising, with the goal of optimizing either welfare or revenue. In the two-stage auction design, a lightweight but coarse model is used in the first stage to quickly select a subset of advertisers to enter the second stage, where a refined but computationally heavier model is used to determine the final winners.

Pros:

1. The paper presents a well-structured and technically sound approach to the two-stage auction design problem in online advertising. The theoretical analysis and derivation of the selection metrics are rigorous and provide strong theoretical guarantees.

2. The paper is well-written and easy to follow. The problem formulation, methodology, and experimental evaluation are clearly explained.

3. The proposed selection metrics, Max-Wel and Max-Rev, are novel contributions that directly address the optimization objectives of welfare and revenue maximization, respectively. This is an important advancement over existing work that primarily focuses on welfare optimization.

4. The two-stage auction design problem is a crucial challenge in modern online advertising systems, where scalability and efficiency are critical. The authors' approach provides a principled solution that can be readily applied in practice.

5. The experimental evaluation is conducted on a synthetic dataset based on IPinYou (which really is the only publicly available dataset for this purpose) and the industrial dataset.

Cons:

1. The paper makes some simplifying assumptions, such as the independence of bids and partial features across ads. While these assumptions enable theoretical analysis, they may not always hold in real-world scenarios.

2. The paper compares the proposed methods to several baselines, it would be valuable to benchmark against more recent, state-of-the-art approaches.

**Questions:**

How sensitive are the proposed metrics to the accuracy of the coarse CTR model used in the first stage?

**Reviewer Confidence:**

2: The reviewer is willing to defend the evaluation, but it is likely that the reviewer did not understand parts of the paper

**Scope:**

3: The work is somewhat relevant to the Web and to the track, and is of narrow interest to a sub-community

---

### Official Review · Reviewer_CPBp · 2024-12-01

**Novelty:** 5
**Technical Quality:** 1

**Review:**

## Summary

The authors consider selling $K$ spots to $n$ advertisers. Having exact knowledge of click-through rate (CTR) makes this problem easy by running the GSP auction. However, when CTRs have to be calculated using some slow machine learning model, it might not be feasible to calculate it for all advertisers. Therefore, the authors consider the problem of selecting a subset of the $n$ advertisers of size $m$ using a less accurate model, in order to perform the full estimation on this reduced problem. They focus on welfare or the revenue maximization.

The authors define a score for each advertiser that they use to select the smaller subset. Their main theoretical results are lower bounds on $m$ in order to guarantee a $\alpha$ approximation of welfare/revenue of the auction when run with full information of the CTRs. The simplest form of their result is for welfare maximization when these scores are uniform random variables in the interval $[a, b]$. In this case, if $n$ is large and $m > \frac{b}{b-a} (1 - \sqrt{1-\alpha}) n $, then their algorithm is guaranteed an $\alpha$ approximation of the welfare with exact CRTs. For general distributions with standard deviation to mean ratio $\rho$ (i.e. $\rho = \sigma/\mu$ where $\mu$ is the mean and $\sigma^2$ is the variance), they show that $m > n (\alpha + \sqrt{2 \rho}/2 - \rho/2)$ leads to $\alpha$ approximation. They also provide similar bounds for revenue that are much more complicated. Finally, they include experiments that show their method is better than other regression-based methods.


## Evaluation

I really like the problem that the authors are considering. I also think that defining the smaller problem given the ranking indices makes sense and should lead to better results than simpler methods, as the authors' experiments show.

However, I am not convinced by the authors' theoretical results on how large the smaller problem should be. Consider the simple case for welfare maximization and uniform distribution when $b = 1$ and $a = 0$, Lemma 2 (the first bound on $m$ I wrote above). This bound seems to be most useful when $\alpha$ is close to $1$, in which case the required bound becomes that the authors require the CTRs of almost all advertisers. In contrast, when $\alpha$ is small, then the bound is very close to $\alpha n$ which I think is the bound on the welfare if you select a random set of advertisers of size $\alpha n$. In addition, what seems puzzling to me is that the bound is not invariant to translations of the distribution, i.e., moving the distribution up or down.

Lemma 3 (the second bound I have above) is even more puzzling. The bound on $m$ becomes non-decreasing and negative when $\rho$ becomes big enough (i.e. when the standard deviation becomes quite bigger than the mean). Specifically, when $\rho \ge 0.5$, the bound becomes decreasing in $\rho$ and when $\rho \ge 5.3$, the bound becomes negative even if $\alpha = 1$. Unless I have made some major mistake or there is a limit on the values $\rho$ is allowed to take, this seems like a major problem.

What also worries me is that these two Lemmas are stated without any comments on their results or comparison of the two bounds. Similar problems apply to Lemmas 6 and 7.

These observations are very unfortunate, because, as I mentioned before, I think the model (and the solution) that the authors propose are interesting.


## Minor errors

* Lines 272-273: not sure if you can assume that advertisers are value maximizers since in the case GSP might not be incentive compatible.
* In the equations of lines 250-255 you need to have a restriction on the size of $M$.
* In both the proofs of Theorems 1 and 3, I do not know why you mention the first display inequality in each theorem, e.g. equation (6) in Theorem 3. What does it add to the proof?

**Questions:**

Please comment on the questions about Lemmas 2 and 3.

**Reviewer Confidence:**

3: The reviewer is confident but not certain that the evaluation is correct

**Scope:**

4: The work is relevant to the Web and to the track, and is of broad interest to the community

---

### Official Review · Reviewer_4Kqq · 2024-12-03

**Novelty:** 3
**Technical Quality:** 6

**Review:**

The authors propose a prior-free method to solve the two-stage auction design problem, where the two stages are designed for scalability of large-scale internet ad auctions. The settings is a "low computation cost" approximate first stage using less accurate predictions and a "high computation cost" second stage using the most accurate predictions available.

The authors describe the key contributions and theorems clearly. The original contributions over existing work on two-stage auction design are
(a) theoretical guarantees
(b) incorporating maximum revenue as an additional objective

Pros:

The key idea is to learn the relationship between the approximate first stage and accurate second stage predictions directly from data, and then use this to adjust the first stage decisions. With this information, the first stage auction mechanism simplifies to a greedy mechanism.

Cons:

The work does not consider the main reason behind having the two-stage approach in the first place: computational cost. Incorporating the computational cost metric as part of the optimization objective will add a unique novel element to this work, and make this a much stronger contribution to literature. In other words, this is the problem of selecting the parameter `m` for the first stage.

**Questions:**

1. Can the authors please provide citations for the three points describing how existing work falls short?
2. Can the authors please describe the reasoning behind choosing the three baseline methods and why they are appropriate baselines for the proposed mechanism?

**Reviewer Confidence:**

4: The reviewer is certain that the evaluation is correct and very familiar with the relevant literature

**Scope:**

4: The work is relevant to the Web and to the track, and is of broad interest to the community